# Overcoming Barriers in Cancer Biology Research: Current Limitations and Solutions

**DOI:** 10.3390/cancers17132102

**Published:** 2025-06-23

**Authors:** Giovanni Colonna

**Affiliations:** Medical Informatics Unit–AOU L. Vanvitelli, Università della Campania, 80138 Naples, Italy; giovanni.colonna@unicampania.it

**Keywords:** cancer, novel approaches in studying cancer, tumor heterogeneity, tumor microenvironment, drug resistance, determinism and indeterminism in cancer, deep molecular mechanisms in cancer, advanced approaches for cancer progression, cancer stem cells, hereditary malignancies, epigenetic heterogeneity

## Abstract

Cancer research faces significant biological, technological, and systemic limitations that hinder the development of effective therapies and improved patient outcomes. Traditional preclinical models often fail to accurately replicate the complex architecture, microenvironment, and immune interactions present in human tumors. This discrepancy results in promising laboratory findings, not translating effectively into clinical success. A core obstacle is hereditary malignancies and cancer stem cells. They induce tumor heterogeneity, characterized by genetic, epigenetic, and phenotypic variations within tumors, complicating treatment strategies and contributing to drug resistance. The biological processes driving metastasis are only partially understood, limiting therapeutic advances. Beyond biological barriers, systemic challenges include limited funding, regulatory complexities, and disparities in care access across different populations. Overcoming these obstacles requires multidisciplinary collaborations, advanced modeling techniques that better emulate human cancer, and innovative technologies for early detection and targeted therapy. While the cancer complexity and systemic challenges are formidable, ongoing scientific progress and collaborative efforts inspire hope for breakthroughs that can transform cancer diagnosis, treatment, and survival outcomes worldwide.

## 1. Introduction

Cancer remains one of the most formidable challenges in modern medicine, affecting millions globally and posing a significant threat to human health. While decades of dedicated research have yielded remarkable progress in understanding, diagnosing, and treating various forms of this disease, several fundamental limitations continue to impede our ability to win. This review aims to provide a comprehensive overview of the most significant current limitations in cancer research and our biological understanding, drawing upon recent scientific findings and expert perspectives. These limitations span the entire spectrum of cancer research, from the initial stages of preclinical modeling to the complexities of clinical translation, the intricacies of tumor heterogeneity, the quest for early detection, the influence of the tumor microenvironment, the challenges of selective targeting, the pervasive issue of drug resistance, and the elusive nature of metastasis [1]. Addressing these multifaceted barriers is crucial for accelerating progress and ultimately improving outcomes for individuals affected by cancer.

In cancer research, accurately modeling the disease in the laboratory setting is one of the major hurdles. Traditional preclinical models, such as two-dimensional (2D) and three-dimensional (3D) cell cultures and murine xenograft models, have long served as the cornerstone of initial drug testing and biological investigation [2]. However, these systems often cannot capture the intricate complexity of human cancers, leading to a significant disconnect between promising preclinical findings and disappointing results in clinical trials [1]. For instance, 2D cell cultures lack the complex three-dimensional architecture, the crucial cell–cell and cell–matrix interactions, and the diverse cellular composition that characterize tumors within the human body [3]. While 3D cell cultures offer some improvements in structural organization, they often still lack vital components such as a functional vascular network, a comprehensive tumor microenvironment, and a competent human immune response [4]. Murine xenograft models, which involve implanting human cancer cells into immunocompromised mice, also present substantial limitations [5]. The absence or impairment of a fully functional immune system in these mice cannot account for the critical role the immune system plays in both cancer development and the response to immunotherapies. Cancer cell lines may not metastasize or might exhibit altered responses when grown in a murine microenvironment, which differs significantly from the human milieu [6]. Even patient-derived xenografts (PDXs), while better at maintaining the original tumor’s histology and genomic characteristics, often lose human stromal components like fibroblasts, which are replaced by murine counterparts, further distorting the tumor microenvironment [7]. The exceptionally high failure rate of new cancer drugs in clinical trials—approximately 95% never reach the market despite promising lab results—stems, experts believe, from these fundamental inadequacies in traditional preclinical models. Difference in opinion highlights a critical need for more sophisticated and physiologically relevant preclinical systems that can better predict how potential therapies will behave in human patients.

## 2. Genetically Engineered Models

Recognizing the limitations of traditional models, researchers are actively pursuing more advanced preclinical systems to better mimic human cancer biology. Scientists engraft human cells, tissues, or even a human immune system into immunodeficient mice to create humanized mice [8]. These models offer potential advantages, such as the ability to test therapies that are not yet approved for human use and a more relevant immune context for evaluating immunotherapies. However, even humanized mice have limitations, including the incomplete reconstitution of the human immune system, the considerable cost and technical complexity involved in their generation and maintenance, and that their underlying murine physiology still differs from that of humans [8]. Patient-derived xenografts (PDXs) represent another step towards improved modeling; in this process, researchers implant tumor tissue directly from a patient into immunodeficient mice [9]. While PDXs can preserve the original tumor’s histological and genomic features, they suffer from the lack of a fully intact human tumor microenvironment, as murine ones often replace human stromal components [2,9]. The expense and difficulty of generating PDXs, along with their limited applicability for large-scale drug screens, restrict their widespread use. Organoids, three-dimensional in vitro cultures derived from patient tumor tissues or stem cells, have emerged as a promising avenue for preclinical research [10]. These models can recapitulate some aspects of tissue architecture and cellular heterogeneity, offering a better representation of human cancer compared to traditional cell lines [10,11]. However, current organoid systems often lack a fully developed vascular system, a complete tumor microenvironment with all its cellular and acellular components, and standardized culture protocols, which are needed to ensure reproducibility across different labs. Genetically engineered mouse models (GEMMs), in which researchers genetically modify mice to develop cancer, are valuable for studying cancer development driven by specific genetic mutations [11,12]. However, they may not always accurately predict drug responses in humans because of species-specific differences in pharmacology and safety, and their generation and maintenance can be time-consuming and expensive [13,14,15]. The ongoing evolution of preclinical models, from simple cell lines to sophisticated organoid systems and humanized animals, reflects a growing understanding of the need to capture the complexity of human cancer more accurately. However, that each model system still presents its own set of limitations underscores the significant scientific challenge of perfectly mimicking the intricate biology of this disease in a controlled laboratory setting. This causes a careful consideration of the strengths and weaknesses of each model in relation to the specific research question and often requires a multi-model approach for a more comprehensive understanding.

## 3. Tumor Heterogeneity

A major limitation that pervades all aspects of cancer research and treatment is the inherent heterogeneity of tumors. Cancer is not a monolithic disease but a complex collection of diverse cell populations that exhibit significant variations at the genetic, epigenetic, and phenotypic levels [16]. Genetic heterogeneity refers to the variations in the DNA sequence among different cancer cells within the same tumor (intra-tumoral heterogeneity) or between tumors from different patients with the same cancer type (inter-tumoral heterogeneity) [16,17]. This genetic diversity arises from the accumulation of mutations, genomic instability (such as increased mutation rates and chromosomal abnormalities), and exposure to environmental mutagens [16,17]. Epigenetic heterogeneity describes the differences in gene expression patterns that occur with no changes to the underlying DNA sequence. These variations in gene expression can be driven by factors such as DNA methylation and histone modifications, leading to different cellular phenotypes even among genetically identical cells. Phenotypic heterogeneity refers to the resulting variations in observable characteristics and functional behaviors of cancer cells, including differences in morphology, proliferation rate, metabolism, drug sensitivity, and metastatic potential [18]. This heterogeneity is not a static feature but a dynamic process, with different subclones of cancer cells constantly evolving and adapting over time and in response to various selective pressures, including anti-cancer treatments [19]. The multi-layered and dynamic nature of tumor heterogeneity represents a profound challenge to our understanding and treatment of cancer. Section 14.3 will provide more details.

## 4. Effectiveness of Cancer Therapies

The existence of this intricate cellular diversity within and between tumors has significant implications for the effectiveness of cancer therapies, particularly targeted therapies and immunotherapy [19]. Intra-tumoral heterogeneity is a major driver of both intrinsic and acquired resistance to targeted therapies. If a targeted therapy focuses on a specific mutation or pathway present in only a subset of cancer cells, those cells lacking the target will be inherently resistant and can eventually proliferate to repopulate the tumor [20]. The selective pressure exerted by the therapy can lead to the outgrowth of minor, resistant subclones that were present from the beginning or arise during treatment because of ongoing genetic and epigenetic evolution [19]. Tumor heterogeneity also poses considerable challenges for the effectiveness of immunotherapy [21]. Immunotherapies often rely on the immune system, recognizing specific tumor-associated antigens. However, heterogeneous tumors may contain subpopulations of cells that express these antigens at different levels or even lack them entirely, leading to immune evasion [21].

Different subclones within a tumor may display varying immunogenicity and express immune-suppressing molecules, thus creating an immunosuppressive microenvironment that impairs the ability of immune cells to target and eliminate cancer [22]. Tumor heterogeneity also complicates the diagnostic process. A single biopsy, often used to guide treatment decisions, may not capture the full spectrum of genetic and molecular alterations present throughout the entire tumor or in distant metastatic sites, potentially leading to inaccurate treatment selection [23,24]. Overcoming the challenges posed by tumor heterogeneity requires the development of novel therapeutic strategies that can simultaneously target multiple vulnerabilities across different cell populations within a tumor. This may involve the use of combination therapies, adaptive treatment approaches guided by real-time monitoring of tumor evolution, and the exploration of therapies that target the underlying mechanisms driving heterogeneity itself [24]. Single-cell sequencing and spatial genomics technologies offer unprecedented opportunities to characterize and understand tumor heterogeneity at an unprecedented level of detail, paving the way for more personalized and effective treatment strategies [24]. Table 1 shows some effects of tumor heterogeneity on cancer therapy.

## 5. Early Cancer Detection

Experts widely recognize early cancer detection as critical for significantly improving patient survival rates and treatment outcomes because early-stage cancers are more localized and amenable to curative therapies [25,26]. However, currently available cancer screening methods for various cancer types face significant limitations. Imaging techniques such as mammography, CT scans, and MRI, while valuable for detecting morphological changes, often lack the sensitivity to detect slight, early-stage tumors and may not always be inherently cancer-specific, leading to false positives [27,28]. Their accessibility and cost can be limiting factors for widespread screening [27,28]. Many established tumor markers, such as PSA and CA-125, have showed poor accuracy and efficacy, particularly for screening prevalent cancers. The low sensitivity and specificity of these markers lead to false positives and false negatives. Non-cancerous conditions may also elevate their levels.

Invasive procedures like colonoscopy and biopsy, while crucial for definitive diagnosis, can be uncomfortable, carry inherent risks, and may not be suitable for widespread screening of asymptomatic individuals [29,30,31]. The challenge of overdiagnosis, where screening detects cancers that would never have progressed to cause symptoms or death, leading to unnecessary anxiety and treatment, also remains a concern [29,30,31]. Factors like lead-time bias and length bias can complicate the evaluation of screening effectiveness, where earlier diagnosis through screening may not translate to improved overall survival [32]. Significant disparities in cancer screening rates and outcomes persist across different population groups, often linked to socio-economic factors, access to healthcare, cultural beliefs, and mistrust of the healthcare system [33]. These limitations highlight the ongoing need for more accurate, less invasive, and more accessible early detection strategies that can benefit all populations equally. The development of multi-cancer early detection (MCED) tests, which aim to detect multiple cancer types through a single blood test, represents a promising but still investigational approach to overcome some of these limitations [34]. However, widespread implementation requires rigorous validation in large clinical trials to show their effectiveness in reducing cancer mortality and carefully assess the risks of false positives and overdiagnosis. Table 2 shows limitations and challenges in early cancer detection.

## 6. Emerging Technologies

Emerging technologies hold considerable potential for overcoming the limitations of current early detection methods. Multi-omics approaches, integrating data from genomics, transcriptomics, proteomics, and metabolomics, may provide a more comprehensive and sensitive way to detect early cancer signals [35]. However, challenges in data integration, interpretation, and standardization, as well as ethical considerations regarding data privacy, need to be addressed [35]. Nanotechnology offers the potential to develop novel molecular contrast agents for imaging and highly sensitive in vitro assays for detecting circulating tumor cells (CTCs) and other early cancer biomarkers [36,37]. However, translating these technologies from the laboratory to clinical practice and ensuring their safety and efficacy remain key challenges [36,37]. Artificial intelligence (AI) and machine learning can analyze medical images and other data to detect subtle signs of early cancer that the human eye may miss and to integrate various data types to improve diagnostic accuracy [38,39]. Challenges include ensuring data quality and security, verifying algorithm reliability and transparency, integrating AI into existing healthcare systems, managing implementation costs, and addressing ethical and regulatory considerations [40,41,42]. Liquid biopsies, which involve analyzing blood or other bodily fluids for cancer-derived molecules, offer a less invasive approach to early detection and monitoring [43]. However, the low concentration of these analytes in early-stage cancers and the need for highly sensitive and specific detection methods remain significant hurdles [43,44]. While these emerging technologies offer exciting possibilities for improving early cancer detection, their successful translation into clinical practice requires overcoming significant technical hurdles, ensuring clinical validity and utility, and pondering ethical and regulatory implications. A collaborative effort involving researchers, clinicians, technology developers, regulatory agencies, and patient advocates is essential to navigate these challenges and realize the full potential of these innovations to affect cancer outcomes. The limitations of traditional preclinical cancer models and emerging alternatives are shown in Table 3.

## 7. Tumor Microenvironment

The tumor microenvironment (TME), the complex ecosystem surrounding cancer cells, plays a critical role in all stages of cancer development and treatment response [45]. This dynamic environment comprises various non-cancerous cells, including stromal cells (like fibroblasts), endothelial cells forming blood vessels, and immune cells, as well as the extracellular matrix (ECM), soluble factors (like growth factors and cytokines), and the local physical conditions (such as oxygen and nutrient levels) [45,46]. The TME can provide essential growth signals, nutrients, and physical support that promote cancer cell proliferation and survival [47,48]. It is also crucial for angiogenesis, the formation of new blood vessels that supply tumors with oxygen and nutrients [49]. Interactions with the TME can facilitate cancer cell invasion into surrounding tissues and metastasis to distant organs, with the ECM often being remodeled to create pathways for cancer cell migration. The TME can contain immune cells that are suppressed or reprogrammed by cancer cells to promote tumor growth rather than attack it, and cancer cells can express molecules that directly inhibit immune cell activity [50]. Notably, the TME can significantly contribute to resistance against various cancer therapies, including chemotherapy, radiation therapy, targeted therapy, and immunotherapy, by providing a protective niche, secreting factors that shield cancer cells from drugs, hindering drug penetration, and creating hypoxic regions that reduce the efficacy of certain treatments [50,51]. The inherent heterogeneity within the TME itself, with different spatial regions exhibiting variations in cellular composition, oxygen tension, nutrient availability, and signaling molecule concentrations, further complicates treatment responses [52]. The tumor microenvironment is not a passive spectator but an active participant in cancer development and treatment response [53], making the understanding of its intricate interactions with cancer cells crucial for developing more effective therapies.

The growing recognition of the TME’s critical role has led to the exploration of therapeutic strategies that target its various components. Immunotherapies aim to modulate the immune cells within the TME, such as T cells, to enhance their anti-tumor activity and overcome immune suppression [54]. Targeting stromal cells, particularly cancer-associated fibroblasts (CAFs), which are a major component of the TME in many cancers and contribute to ECM remodeling and drug resistance, is also being investigated [55]. Anti-angiogenic therapies aim to inhibit the formation of new blood vessels within the TME, starving tumors of oxygen and nutrients [56]. Emerging research also suggests that the gut microbiota can influence the response to certain cancer therapies, particularly immunotherapies, leading to the investigation of strategies to modulate the gut microbiome to enhance treatment efficacy [57]. While targeting the TME offers a diverse range of therapeutic opportunities that complement traditional approaches focused on cancer cells themselves, the complexity and heterogeneity of the TME cause a deeper understanding of its specific roles in different cancer types and stages to develop effective and safe interventions. Advances in spatial multi-omics technologies are providing new ways to map and understand the heterogeneity of the TME at a cellular and molecular level [58,59], offering invaluable information for identifying novel therapeutic targets and developing more precise strategies for modulating the TME in cancer treatment.

## 8. Treatments Selectively Targeting Cancer Cells

A central goal in cancer therapy is to develop treatments that can selectively target cancer cells while minimizing harm to healthy tissues [60], reducing treatment-related toxicities and improving patient quality of life [61]. However, achieving this selectivity is inherently challenging because cancer cells arise from normal cells and share many of their fundamental molecular characteristics and pathways. This makes it difficult to identify therapeutic targets that are only present in or essential for cancer cells. The concept of “undruggable” targets further complicates this issue [62,63]. These are proteins that play critical roles in cancer development and progression but lack suitable binding sites for traditional small-molecule drugs or antibodies, making them inaccessible to therapeutic intervention [64,65]. Even targeted therapies, while designed to focus on specific molecular alterations, can still have off-target effects because of the presence of the target in some normal cells or the involvement of the targeted pathway in normal cellular processes [66]. Traditional cytotoxic chemotherapies, which lack specific targeting mechanisms, often cause significant side effects by damaging rapidly dividing normal cells besides cancer cells [67,68]. The pursuit of highly selective cancer therapies is therefore a continuous endeavor that requires innovative approaches to overcome the inherent similarities between cancer and normal cells and to expand the repertoire of druggable targets.

Researchers are constantly exploring novel approaches to enhance the specificity of cancer therapies. Targeted therapies continue to be refined to focus on more specific molecular alterations that are uniquely or predominantly found in cancer cells, including the development of more potent and selective inhibitors of known oncogenic drivers [63,64,69,70]. Immunotherapy has emerged as a transformative approach that leverages the patient’s own immune system to recognize and selectively destroy cancer cells based on their unique antigens [71]. Advances in understanding tumor immunology have led to the development of various immunotherapeutic strategies, such as immune checkpoint inhibitors and CAR T cell therapy [72,73]. Antibody-drug conjugates (ADCs) represent another strategy to enhance specificity by combining the targeting ability of antibodies that recognize cancer-specific surface antigens with the potent cytotoxic effects of chemotherapy drugs, allowing for targeted delivery directly to cancer cells and reducing systemic toxicity [74,75]. Strategies for protein degradation, such as PROTACs and molecular glues, offer a completely new way to target previously “undruggable” proteins [76]. These strategies use the cell’s own protein degradation machinery for the selective breakdown of these proteins [77]. While these more targeted approaches represent significant advancements, they are not universally effective, and resistance can still develop, underscoring the need for continued research to identify new targets and refine these approaches to improve their efficacy and reduce side effects. Integrating advanced diagnostics, such as genomic profiling and liquid biopsies, is becoming increasingly important for identifying patients who are most likely to benefit from specific targeted therapies and for monitoring treatment response and resistance mechanisms [78,79].

## 9. Drug Resistance

The development of drug resistance, where cancer cells become less responsive or unresponsive to treatment, is a major obstacle to successful long-term cancer therapy and a primary reason numerous cancers recur and progress [80]. Cancer cells can acquire resistance through a variety of complex mechanisms, including genetic mutations in the drug target or downstream signaling pathways, epigenetic modifications that alter gene expression, activation of alternative survival pathways, increased expression of drug efflux pumps that remove the drug from the cell, changes in drug metabolism that inactivate the drug, and interactions with the tumor microenvironment that provide protection [81,82,83]. Notably, intra-tumoral heterogeneity plays a critical role in the development of drug resistance. The presence of diverse subclones within a tumor increases the likelihood that some cells will possess or acquire resistance mechanisms. The selective pressure of the drug then eliminates the sensitive cells, allowing the resistant clones to expand and dominate [84,85]. Researchers believe that cancer stem cells (CSCs), a subpopulation of cancer cells with self-renewal and differentiation capabilities, also contribute to drug resistance and tumor recurrence [86]. The ability of cancer cells to develop and adapt under the selective pressure of therapy is a fundamental limitation in oncology.

To overcome the challenge of drug resistance, researchers are exploring various strategies. Combination therapies that target multiple pathways simultaneously with different drugs can reduce the likelihood of resistance development by making it harder for cancer cells to adapt [87]. Developing drugs that specifically inhibit known resistance mechanisms can restore the sensitivity of cancer cells to the primary therapy [88,89]. Epigenetic therapies, which modify gene expression patterns, have the potential to reverse drug resistance by altering the expression of resistance-conferring genes [90]. Adaptive therapy, which involves adjusting the dose or type of therapy based on the tumor’s response, aims to maintain a balance between drug-sensitive and drug-resistant clones, potentially prolonging the effectiveness of treatment [91]. Liquid biopsies can monitor the emergence of resistance-conferring mutations in real-time, allowing for early detection of resistance and timely switching to alternative therapies [43,44,78,79]. Ultimately, overcoming drug resistance will probably require a multifaceted approach that combines different therapeutic modalities, targets both cancer cells and their microenvironment, and uses sophisticated monitoring strategies to adapt treatment in real-time. International collaborations and data sharing are crucial for accelerating the discovery and development of new therapies and strategies to combat drug resistance in cancer [92].

## 10. Metastasis and Molecular Mechanisms

Metastasis, the spread of cancer cells from the primary tumor to distant sites in the body, is the leading cause of the vast majority of cancer-related deaths [93]. This complex, multi-step process involves a local invasion of surrounding tissues, intravasation into the bloodstream or lymphatic vessels, survival in circulation, extravasation at a distant site, and colonization to form a secondary tumor [93,94]. Despite significant research efforts, there are still considerable gaps in our understanding of the specific molecular mechanisms that govern each step of this metastatic cascade. For instance, we do not fully understand what makes some cancer cells gain the ability to metastasize while others do not, what determines the specific organs to which a particular cancer type will metastasize (metastatic tropism), how metastatic cells survive and thrive in the microenvironments of distant organs, and what early events start the formation of a pre-metastatic niche in distant organs, preparing them for cancer cells. Modeling the metastatic process accurately in preclinical animal models remains a significant challenge because many models, particularly those using cell lines, fail to spontaneously metastasize or metastasize to the same organs as in humans [95,96]. The complexity of metastasis, coupled with our incomplete understanding of its underlying mechanisms, represents a major barrier to improving cancer survival rates.

Researchers are focusing on identifying and targeting key molecules and pathways involved in metastasis. This includes targeting cell migration and invasion by inhibiting molecules that promote cancer cell motility and their ability to degrade the ECM [97,98]. Preventing the formation of new blood vessels (angiogenesis) at metastatic sites, which is necessary for the growth of secondary tumors, is another important strategy [99]. Understanding how cancer cells adapt to the microenvironment of distant organs and developing therapies that interfere with this process are crucial for preventing the formation of secondary tumors. Researchers are also investigating the signals that prepare distant organs for cancer cells and developing therapies that can disrupt the formation of these pre-metastatic niches [100]. Leveraging the immune system to detect and eliminate metastatic cancer cells through immunotherapeutic approaches is another promising avenue [101]. Preventing and treating metastasis requires a multi-pronged approach that targets various stages of the metastatic cascade and involves a combination of therapies aimed at both the cancer cells and their interactions with the microenvironment at both primary and secondary sites [102]. The development of more sophisticated preclinical models that accurately mimic human metastasis and allow for real-time tracking of metastatic spread, along with the identification of early biomarkers of metastatic potential, is crucial for accelerating progress in this critical area of cancer research [103].

## 11. The Transition from Determinism to Indeterminism in Biomedicine

Much of cancer research is based on observational studies that correlate macroscopic aspects [104], such as symptoms or molecular markers, with cancer and its progression. This simplistic perspective has evolved into probabilistic and indeterministic biological systems biology view of the present day [105,106,107], where the interactions between biomolecular populations determine functionality. However, even today, many reductionist approaches [108,109] are quite common [110]. It is a cultural problem that has deep roots in the methodologies of the last century.

Cellular metabolism operates through interactions between proteins, in networks of deep and highly regulated molecular processes that interact functionally with each other [111,112,113]. Regarding protein interactions, proteins often interact briefly via transient interactions with other proteins within a functional module, they may interact over a longer duration to integrate into a protein complex via a permanent interaction (e.g., ribosomes), or they may interact with a protein for transportation. All this means that without knowing where, how, and when the protein interacts, it is impossible to define its real biological role. The concept of coordinated interaction between genes and/or proteins is fundamental [114,115,116]. Therefore, we can identify entire regulatory networks in which genes and proteins work together in a coordinated manner to promote tumor growth and survival. Some of them are crucial nodes that modulate multiple cellular pathways, even simultaneously, and therefore may represent interesting therapeutic targets, but it is unlikely that a single gene suffices to block disease progression [117,118,119]. For example, in pancreatic cancer, recent studies have gone beyond identifying single mutations. They analyzed gene expression patterns and epigenetic modifications to understand how the cellular environment influences tumor malignancy [120,121,122]. This approach helps to develop combination therapies that target entire molecular circuits, rather than relying on a single target, reducing the risk of drug resistance. In actuality, biomolecules communicate through interactions and functional relationships [123]. Biological functions have a very complex informational origin [124]. They can arise from the interaction of single molecules that exchange information from the outside, which is then communicated to the set or group of molecules to which they belong and with which they develop common activities [125]. A typical example used for information transfer in cells is the well-known physical and functional interaction between proteins. Therefore, it is a group of biomolecules, not a single biomolecule, that performs any biological function [126,127]. By interacting with each other [128,129], they exchange data/information (elementary information event) that is then mediated by the entire group or relational system (subgraph or functional module) [130,131,132]. The common relational activity leads to the emergence of a functional property, characteristic of that subgraph/functional module [133,134]. These biological events are informational because the function is the element that derives from the joint processing of elementary events of physical interaction (or even sequences of elementary acts (or bits) of analog communication), whether long-lasting (e.g., in complexes) or momentary, transmitted through a very complex interactive network (digital communication) that processes them producing a biological function (the meaning of the processing) [135,136]. Therefore, inter-cellular and intra-cellular metabolic relationships are tightly connected to implement specific and common functional purposes for a tissue (or organ) (implementation of informed decisions) by nonlinear dynamics [137,138]. We can consider their behavior similar to that of an internet network, with nodes and hubs and a modular organization of subgraphs with organizing centers often organized according to hub–spoke patterns [139,140,141]. The organization of the organism’s circadian homeostasis critically depends on this complex informational network [142,143]. Homeostasis expresses the robustness in responding to internal metabolic variations to maintain metabolism in its fundamental state while also responding quickly and effectively to external perturbations, be they metabolic or environmental. The organizational model with hub–spoke centers guarantees both robustness and adaptation, where the HUB center concentrates and regulates, through its network of nodes (spoke centers), many functional “services” [144,145,146]. This pattern is common to very different sectors of human activity, such as urban planning, flight control, or social systems [140]. Specific pathways (e.g., signaling pathways) direct information flow to ensure rapid response and information security within such a complex system [147,148]. Sequential interaction between component nodes (proteins), exchanging elementary bits of information to aggregate and form complex data, defines the physical support for signaling pathways [149,150]. The hub and bottleneck nodes (proteins) represent the regulatory components of the flows and the crossing points between different flows that we can calculate from the topological parameters of the network [150]. As in IT, the bit is the standard unit for measuring information, defining the quantity of biological information as entropy (Shannon entropy) [151,152,153,154]. These flows define deep molecular processes or mechanisms, and they are the ones that represent the cause of what happens at the macroscopic observational level. At these levels, within the realm of biomolecules, governing processes are subject to indeterministic physical laws, which are not linearly correlated with the macroscopic world’s classical cause-and-effect principles [155,156,157]. This also means that, when we identify a tumor marker, we must also characterize it, demonstrating to which specific deep functional sub-network it belongs. The lack of this characterization renders all statements and their opposites statistically equivalent, accounting for the lack of efficacy or presence of side effects in some anti-cancer drug molecules despite successful target binding.

## 12. The Importance of Deep Molecular Mechanisms in the Study of Cancer

The study of the deep molecular mechanisms of cancer is fundamental to understanding its onset, progression, and resistance to treatments [158,159]. In recent years, research has made progress in identifying genetic mutations and metabolic alterations that influence tumor growth. Metabolic alterations focus on the metabolic plasticity of tumor cells, which represents the ability to adapt to environmental conditions and develop drug resistance. For example, the EU-funded CANCER METASTASIS project (ADAPTMET) aims to address metastasis from four key scientific angles (cell fate, environment, latency, and expansion) in order to identify key genes involved in cancer metastasis. It is a multidisciplinary approach whose results could lead to the development of new targeted treatments to counteract the spread of tumors.

However, as we have noted, no gene acts in isolation [160,161]. Each gene is part of complex molecular networks that interact dynamically during cancer progression. The concept of a “key gene”, which is often used to simplify scientific communication, can be misleading if not correctly contextualized. We often read that “cancer arises from a mutated gene”. We also use this phrase to describe the cause of the disease. Despite its general usage, this phrase might lead to a flawed understanding of genetics. It expresses a reductive, deterministic view, which equates an individual’s entire genetic code with a single individual trait, as if we could also have another single gene that makes us immune to cancer. The growing use of a single gene as a tool for understanding causal aspects of a disease supports a deterministic thinking that suggests an immutable genetic makeup operating through single disease-causing genes. After all, even the native protein, the one decoded by the gene, does not exist in metabolic reality.

Almost always, the protein passes through the Golgi/ER system after expression. This system chemically modifies and tailors it to a specific cellular location, where it interacts with other biomolecules to create a function [162]. Thus, although the gene always expresses the same native protein, the different covalent modifications generate many and various proteoforms [163], each of which is adapted to specific molecular relationships in nonidentical cellular locations. Each type of proteoform is a chemically distinct entity from the others, with its own highly specific chemical/physical characteristics. This requires researchers to identify the entire regulatory network in which the specific proteoform works in coordination with other proteins to promote tumor growth and survival [164,165]. Proteoforms are the different forms of a protein produced by the genome through sequence variations, splicing isoforms, and post-translational modifications. They are modified to perform a specific activity in a specific cellular environment and at a specific time. Although the theoretically calculable number of proteoforms is astronomical (>10^27^), a limit to this complexity is the copy number present in cells, which is rather limited, and the number of genes simultaneously expressed in cells [163]. However, the question of how many total proteoforms exist is very difficult to answer [163]. Some estimates indicate around 60,000 characterized proteoforms. But this is still an extremely small number, which today makes this knowledge of little use for broad analyses. This is also because their properties are collapsed on those of native proteins, and there is no specific archive. It does not escape the reader that in these contexts, the characterization of a proteoform is crucial to understanding what its specific molecular properties are. Some nodes are crucial because they intersect and modulate multiple cellular pathways, and therefore become interesting therapeutic targets, but a single gene is rarely enough to block disease progression [166,167]. For example, analyzing gene expression patterns helps to prepare therapies that target molecular pathways that often intersect through a common component. This means targeting the ability of tumor cells to adapt their metabolism in response to environmental conditions at multiple points. As mentioned above, this approach closely relates to cancer progression and treatment resistance.

In addition, cancer cells must face hostile, hypoxic, nutrient-poor environments [168]. To survive, they modify their metabolism in different ways by regulating a series of deep molecular mechanisms that involve the reprogramming of cellular metabolism and the interaction with the tumor microenvironment [169,170]. For example, alterations in AMPK and mTOR signaling, increased expression of glucose transporters (GLUT1 and GLUT3), increased activity of aerobic glycolysis, with the strong production of lactate favors tumor growth [171,172], or even the use of glutamine for the synthesis of nucleotides and the production of NADPH, essential for resistance to oxidative stress [173,174], are part of this picture. Metabolic plasticity is, therefore, also a key element in cancer biology and may represent a target for new therapies. However, epigenetic modifications, such as DNA methylation and histone modification, influence and regulate the expression of genes involved in metabolism; we also cannot exclude their interaction with the tumor microenvironment [175,176,177]. Tumor cells also communicate with fibroblasts and immune cells to modulate local metabolism [178]. For example, they can induce tumor-associated fibroblasts (CAFs) to produce lactate, which is then used by tumor cells as an energy source. The overall picture that emerges is that the combination of these mechanisms makes tumor metabolism extremely adaptable, contributing to treatment resistance and disease progression.

However, we note that cancer molecular mechanisms influence the human mechanisms, i.e., the specific human phenotype that acts as a filter in that specific context [179,180]. The result depends on the filtering capacity of the phenotype, i.e., the molecular mechanisms of contrast that it will use against cancerous ones. This means that, where a specific cancer always produces the same molecular activity to attack, the phenotype counteracts and modifies it. Cancer is not a static disease but an evolutionary phenomenon that interacts with the host phenotype, creating a complex dynamic between tumor molecular mechanisms and the body’s defense mechanisms [180,181,182]. Although there are common oncogenic pathways, such as the activation of oncogenes (e.g., RAS and MYC) and the dysfunction of tumor suppressor genes (e.g., TP53 and RB1), cancer develops continuously through somatic mutations, metabolic plasticity, and interactions with the tumor microenvironment [183], as mentioned above. But this means that even if a tumor starts with a series of initial alterations, its progression can vary based on the biological context in which it develops. In order to evolve, cancer must overcome phenotypic filtering [184,185]. The immune system and the patient’s cellular repair mechanisms act as evolutionary barriers, forcing the tumor to develop new strategies to survive [186], for example, by selecting helpful mutations, or by creating a highly adaptable population within the same tumor, as cell clones with different mutations can coexist within the cancerous tissue.

This means that we cannot consider cancer a single entity; rather, it is a series of diseases adapting to the host’s phenotype [187,188,189]. This also explains why the same type of tumor can have different responses to treatments in different patients. Its evolution is driven by selective pressures, just as occurs in natural evolutionary processes [190].

Lung cancer’s genetic heterogeneity is well known. This leads to clonal selection, where the most drug-resistant cells proliferate, making the tumor more aggressive. Or, in breast cancer, some tumor cells develop resistance to hormone therapy through mutations in estrogen receptors. This phenomenon is an example of evolutionary adaptation, in which the tumor changes its biology to survive therapies. Melanoma is a highly adaptable tumor. Cancer cells can change their gene expression profile, moving from a proliferative to an invasive state, allowing them to metastasize more easily. This type of plasticity is an obvious example of tumor evolution. These examples show that cancer is not a static disease, but a biological system that is constantly developing to promote invasion [191,192]. Appendix A explains the biological meaning of macroscopic and microscopic levels.

## 13. Advanced Approaches to Study Cancer Progression

To follow the progression of cancer and its interaction with human molecular mechanisms, we need dynamic techniques to analyze changes in real time. Some innovative approaches that provide crucial information on its deep mechanisms are as follows.

### 13.1. Cancer Genomics

*DNA Sequencing (Next-Generation Sequencing—NGS)* [193]: Allows for reading the entire DNA sequence of the tumor genome. It allows for the identification of point mutations, insertions/deletions, structural rearrangements, and copy number alterations (gene amplifications or deletions) that are the basis of cancer development. Technologies such as whole-genome sequencing (WGS), whole-exome sequencing (WES), or panels of specific genes (panel sequencing) are fundamental for identifying driver genetic alterations.

*Single-nucleotide sequencing (SNP array)* [194]: Useful for the identification of large-scale copy number variations and loss of heterozygosity.

### 13.2. Transcriptomics

*RNA sequencing (RNA-Seq)* [195]: A measure of the expression of all genes in a sample (tumor or normal). It allows the identification of over- or under-expressed genes in cancer, gene fusions at the RNA level, alternative splicing, and the expression of non-coding RNAs. It provides crucial information on altered transcriptional programs in tumor cells.

*Gene expression microarray* [196]: An older technology but still used to measure the expression levels of thousands of genes simultaneously.

### 13.3. Proteomics

-*Mass spectrometry (MS)* [197]: Allows the identification and quantification of proteins present in a sample. Comparative proteomics of tumor and normal tissues identifies proteins with altered expression or post-translational modification in cancer. It is a fundamental approach to study the actual quantity and functional state of the molecules that perform most cellular processes. It also includes an analysis of post-translational modifications (such as phosphorylation), which are crucial for regulating protein activity.-*Protein arrays* [198]: Similar to DNA microarrays, allow analysis of the expression or activity of many proteins at once.

### 13.4. Epigenomics

-*Bisulphite DNA sequencing (BS-Seq) and derivatives* [199]: Studies DNA methylation patterns, an epigenetic modification that can alter gene expression without changing the DNA sequence. Cancer profoundly alters methylation patterns.-*CHiP sequencing (CHiP-Seq)* [200]: Identifies protein binding sites on DNA, such as transcription factors or histone modifications. Alterations in chromatin structure and protein binding to DNA are common in cancer and affect gene expression.-*ATAC-Seq* [201]: Measures chromatin accessibility, showing regions of the genome that are actively transcribed or regulated.

### 13.5. Single-Cell Technologies

-*Single-Cell DNA/RNA Sequencing* [202,203]: Allows analysis of the genomic or transcriptomic profile of single cells within a heterogeneous population. This is crucial in cancer in understanding tumor heterogeneity, identifying subpopulations of cells with unique characteristics, and studying their evolution and interaction.

### 13.6. Advanced Imaging

-*Super-resolution microscopy and live imaging* [204]: Allows the visualization of molecules and their interactions within cells in unprecedented detail and the study of dynamic processes in real time.-*Mass Spectrometry Imaging* [205]: Allows the determination of the spatial distribution of molecules (proteins, lipids, and metabolites) within a tissue sample.

### 13.7. Bioinformatics and Computational Biology

These are not “wet” experimental technologies, but they are essential for analyzing, interpreting, and integrating the massive amount of data generated by the above technologies. Computational models and algorithms are used to identify patterns, build molecular networks (such as interactomes), predict gene function, simulate cellular behavior, and identify potential therapeutic targets [206]. Therefore, at the microscopic level of cellular metabolism, the deep metabolism, where protein interactions manage and generate biological functions, experimental data relating to these interactions represent the only condition through which we can verify the real existence of a biological function through computational analysis [207,208,209]. It also represents the level where all causal molecular mechanisms act; their effects, not linearly correlated, appear at the higher, macroscopic observational level. The symptoms of a disease or the biological markers we measure in the blood are the visible and measurable effects of what happens below. Thus, computational models become scientifically true only if based on experimental data or on something that allows their interpretation in experimental terms. The experimental validation of information acquired indirectly (for example, only through bioinformatic analyses) is essential to have correct and real functional information [210] and not at the stage of the hypothesis. Unfortunately, despite the identification of tens of thousands of functional relationships between proteins in the human proteome, researchers have experimentally validated less than 10% [211]. This makes it very complex and difficult, if not prohibitive, to establish, with certainty, the reliability of a metabolic model, be it normal or pathological. This experimental “impasse” represents one of the major obstacles in cancer research.

### 13.8. Mathematical Modeling and Stochastic Control

Random and unpredictable factors influence the evolution of cancer. For this reason, scientists are developing stochastic mathematical models that simulate tumor growth and response to treatments, improving the ability to predict the evolution of the disease [212].

### 13.9. Artificial Intelligence and Big Data

Using AI and machine learning can revolutionize oncology research. Advanced algorithms analyze huge amounts of genetic, epigenetic, and metabolic data to identify cancer evolutionary patterns and can suggest personalized therapeutic strategies [213,214,215]. Each of these approaches provides a piece of the puzzle to understand the complex molecular mechanisms of cancer. Scientists believe that integrating data from different platforms (a “multi-omics approach”) is the most effective strategy to acquire a comprehensive and in-depth view of the disease. Interactomics plays a key role in this integration, providing the “wiring” that connects the different molecular components and allows us to understand how alterations at one level (e.g., gene mutation) translate into changes at the protein and pathway levels, ultimately influencing cellular behavior and tumor progression [216,217].

These approaches, and others, are transforming the understanding of cancer and opening new avenues for more effective therapies. But almost all these advanced approaches require “cognitive” interventions on biomedical big data banks. Current data banks are years old and contain heterogeneous data, making it impossible to distinguish reliable from unreliable data [218,219,220]. Under these conditions, no advanced technique will give reliable results. These problems of heterogeneity, obsolescence, and data quality can significantly affect the reliability of analyses based on advanced techniques.

For example, many databases contain information collected years ago, which may not reflect the current genetic and epigenetic landscape of tumors. The data comes from studies with different methodologies, different analytical tools, and heterogeneous populations, making consistent integration difficult. Almost always, the patient cohorts analyzed are not representative of all human phenotypes, limiting the generalizability of the results [221].

The results are only as good as the data they are based on. If these improve, advanced tools will also be more effective. The creation of new dynamic databases with continuous updates through advanced sequencing tools and real-time data collection [222]; new AI models for filtering heterogeneous data, which distinguish between reliable and noisy information; and multi-omics integration techniques, which combine genomic, epigenetic, metabolic and transcriptomic data, will give a more complete view of tumor plasticity. Without forgetting the standardization of protocols, to avoid inconsistencies in data from different laboratories [223], AI is used to filter noisy data, extract only reliable information [224], and, above all, activate international collaborations, creating homogeneous datasets on different populations.

However, to transcend current limitations rooted in heterogeneous data, integrating quantitative, multi-modal datasets has emerged as a pivotal strategy in advancing cancer research. By combining genomics, transcriptomics, proteomics, metabolomics, and advanced imaging within a unified analytical framework, researchers can develop comprehensive models that capture tumor heterogeneity, evolutionary dynamics, and treatment responses more accurately [225]. Computational approaches, including machine learning and mathematical modeling, enable the synthesis of these complex datasets into predictive tools that can guide personalized therapy decisions [226]. Such integrative efforts not only enhance our biological understanding but also facilitate the development of more precise diagnostic and prognostic strategies, ultimately addressing the systemic and biological barriers that currently limit translation into clinical benefit. I would now like to bring hereditary malignant neoplasms as an example of the need for clinicians and researchers to have a wide range of knowledge. It is necessary to possess that common language, which the ancient Greeks called “Koinè diàlektos” (κοινὴ διάλεκτος), indispensable for understanding what others are doing and for making us understand what we do. We should not overlook that this is often an intrinsic limitation of many biomedical communities, stemming from the extremely rapid development of technologies and the evolution of knowledge.

## 14. Hereditary Malignancies

Hereditary malignancies, with a marked familial component, are tumors that develop because of genetic mutations transmitted from parents to children [227,228]. Hereditary cancers are estimated to occur in 5% to 10% of all cancer cases [229]. However, recent studies have revised this percentage upwards, suggesting that it could be as high as 13.3–17.5% [230]. This increase is because of the increasing use of advanced genetic testing, such as Next-Generation Sequencing (NGS), which allows genetic mutations to be identified more accurately. Therefore, hereditary cancers comprise approximately 5% to 17.5% of all cancer cases. The most common malignancies include lung cancer (2.48 million cases), breast cancer (2.3 million), colorectal cancer (1.9 million), and prostate cancer (1.46 million). If we apply this percentage to the 20 million recent cases of cancer recorded in 2022, we can estimate that between 1 million and 3.5 million of these are cancers with an inherited genetic component. This variability depends on classification criteria and access to genetic testing.

Genetic predisposition to tumors refers to inherited mutations that increase the risk of developing a neoplasm later in life. However, having a genetic mutation does not mean that the disease will manifest. Some genetic mutations increase the risk of developing cancer, but they do not guarantee that the disease will manifest itself [231].

Certain mutations in the BRCA1, BRCA2, TP53, APC, MLH1, and MSH2 genes, among others, can increase the risk of cancers such as breast, ovarian, colon, and thyroid cancers. Parents transmit these mutations, which are present in all the body’s cells from birth. If there are several cases of cancer of the same type or tumors at a young age in a family, there may be a genetic predisposition. There are genetic panels that analyze up to 80 genes related to the most common hereditary cancers, allowing pathogenic mutations to be identified [232]. The most common correlations between mutations and associated tumors are between BRCA1 and BRCA2 and breast, ovarian, prostate, and pancreatic cancer; between APC and colorectal cancer (Familial Adenomatous Polyposis—FAP); between MLH1, MSH2, MSH6, and PMS2 and colorectal and endometrial cancer (Lynch syndrome). Alternatively, TP53 is associated with various tumors, including sarcomas, breast cancer, and brain tumors (Li–Fraumeni syndrome), and numerous other less common cancers [233,234]. All this makes genetic diagnosis important, where genetic tests make it possible to identify mutations and adopt personalized prevention strategies, also through regular check-ups for early diagnosis.

### 14.1. Hereditary Cancers and Screening

People with a genetic predisposition should benefit from more frequent screening for early diagnosis. Therefore, access to genetic testing is crucial not only to identify people at risk but also to adopt targeted prevention strategies [235]. The low diagnosis rate for hereditary cancers prevents many people from accessing targeted therapies. Some therapies are more effective for specific genetic mutations. However, new techniques make it possible to reduce side effects and improve efficacy, adapting the treatment to the genetic characteristics of the patient. Using multiple drugs in synergy can improve the therapeutic response and reduce resistance to treatments. In addition, tumor heterogeneity implies that specific types of tumors (e.g., genetic or epigenetic heterogeneity) require distinct therapeutic adaptations (see below). Therefore, therapeutic approaches for hereditary cancers are being developed, thanks to precision medicine and targeted therapies. Here are some of the most promising strategies in targeted therapies concerning PARP inhibitors [236]. These drugs block DNA repair in cancer cells with BRCA mutations, preventing them from growing. They are effective for breast, ovarian, pancreatic, and prostate cancers. Immunotherapy is also developing, which, by stimulating the immune system, recognizes and attacks cancer cells. It is useful for tumors with microsatellite instability, such as those related to Lynch syndrome. Promising new experimental strategies, including mRNA vaccines, are emerging for melanoma and other cancers. Obviously, research is exploring new biomarkers to identify patients who can benefit from innovative therapies. The goal is to develop targeted treatments with fewer side effects.

### 14.2. Genetic or Molecular Mechanisms That Could Transform a Benign Neoplasm into a Malignant One

Transforming a benign neoplasm into a malignant one is driven by accumulating alterations in genes that regulate cell growth, apoptosis, and DNA repair [237,238]. Somatic mutations, which affect gene expression via microRNAs (miRNAs) and epigenetic modifications, can occur [239]. Loss of the ability to control cell proliferation and local invasiveness, with consequent metastasis, are the characteristic signs of malignant transformation. Table 4 shows the various mechanisms.

The amplification or mutation of oncogenes, which stimulates cell growth, leads to uncontrolled proliferation [240]. The inactivation of tumor suppressor genes, which regulate programmed cell death (apoptosis) and inhibit cell growth, promotes the survival of cancer cells. These mutated genes accelerate the accumulation of genomic mutations, increasing the risk of malignant transformation. Chromosomal abnormalities can activate or inactivate genes, altering the control of cell growth [241]. Instead, viruses can insert their genetic material into the genome of cells, causing mutations and neoplastic transformation [242]. And exposure to carcinogens can damage DNA, causing mutations [243].

Epigenetic modifications and miRNAs are more complex and deserve more detail [244]. miRNAs are small RNA molecules that regulate gene expression downstream of transcription. miRNAs can interact with target mRNAs [245], thus inhibiting protein translation or causing mRNA degradation. This mechanism, known as post-transcriptional regulation, allows miRNAs to modulate the expression of specific genes. Epigenetic modifications are heritable changes in DNA function without alterations in the basic DNA sequence. These changes can affect gene expression by altering the structure of chromatin, the region of DNA that associates with histones. DNA methylation, for example, can inactivate genes, making them less accessible to proteins involved in transcription, while histone modification, such as acetylation and methylation, can affect chromatin accessibility and thus transcription [246]. In addition, there are interactions between miRNAs and epigenetic modifications. For example, some miRNAs can modulate DNA methylation, influencing the transcription of target genes. Epigenetic modifications can also affect miRNA production, creating a feedback loop between these two mechanisms [247].

In summary, miRNAs and epigenetic modifications represent two important mechanisms that regulate gene expression, influencing protein production and cell function. Their interaction contributes to the complexity of gene regulation and its relevance in the pathogenesis of different diseases.

### 14.3. Tumor Heterogeneity

As mentioned, inherited germline alterations are present in every cell of an individual’s body from birth. Once a cancer has developed, there is nothing to prevent a new event (a somatic mutation, a loss of heterozygosity, or an epigenetic modification) from also occurring in the remaining healthy copy of that gene. However, this is a stochastic, unpredictable event in which tumor cells within the same tumor can acquire additional distinct mutations, giving rise to intra-tumoral heterogeneity. This event might shape the evolutionary trajectory of the tumor. For example, a germline mutation in a DNA repair gene (such as BRCA1/2) can lead to genomic instability, which increases the frequency of accumulation of somatic mutations. This accelerated mutation rate can contribute to greater genetic diversity within the tumor, favoring the emergence of multiple subclones. The presence of distinct microenvironmental niches within a tumor, coupled with ongoing genetic instability, can lead to the independent evolution of subclones, further increasing intra-tumor heterogeneity. Therefore, inherited cancer syndromes can manifest with different tumor types and locations, even within the same gene or family (inter-tumor heterogeneity). This also means that a single inherited cancer syndrome can predispose individuals to a broader spectrum of tumors than initially thought (for example, still speaking of BRCA1/2 mutations, they primarily cause breast and ovarian cancers, but they can also increase the risk of other cancers.) and links clinical diagnostic and prognostic assessments to assessments of tumor heterogeneity. Indeed, a single biopsy may not capture the entire molecular diversity of the tumor. This is a diagnostic limitation that represents a major challenge in cancer treatment. The presence of genetically diverse tumor cell populations within a single tumor means some subclones may be resistant to a particular therapy, resulting in treatment failure and recurrence. This requires the development of personalized and often multi-modal treatment strategies. Inherited cancers begin with a fundamental genetic alteration that predisposes to cancer. This initial predisposition serves as a starting point for further somatic mutations and evolutionary processes, which then drive development and contribute significantly to the genetic and phenotypic diversity (heterogeneity) observed within and between tumors. Thus, tumor heterogeneity is a fundamental concept in oncology that describes the diversity of cells within a single tumor (intra-tumor heterogeneity) or between tumors of the same type in different patients (inter-tumor heterogeneity) [248]. This diversity is one of the major challenges in cancer diagnosis and treatment, as it affects the effectiveness of therapies and drug resistance.

We delve into the details of genetic and epigenetic heterogeneity and the implications for therapeutic adaptations.

Genetic heterogeneity

Genetic heterogeneity [249] refers to differences in the DNA sequence between cancer cells [250]. These differences may include the following:-*Point mutations*: Changes in a single DNA base.-*Copy number changes (CNVs):* Increases or decreases in the copy number of DNA segments or entire chromosomes.-*Chromosomal rearrangements:* Translocations, inversions, or deletions of large portions of chromosomes.-*Clonal evolution:* A tumor is not a homogeneous mass but comprises several populations of cells, called subclones, which a common progenitor cell that is produced. The cells become more ruthless and resistant to therapies as they accumulate different genetic mutations during tumor growth. This process of “clonal evolution” can occur both in the primary tumor and in metastases.-*Acquisition of resistance*: Genetic heterogeneity is a key mechanism for developing drug resistance. If a therapy targets a specific genetic alteration present in only a part of the cancer cells, subclones that do not possess that mutation or that have developed alternative mutations can survive and proliferate, leading to disease recurrence.-*Differences between primary tumor and metastases:* Metastases can have a genetic profile that differs from the primary tumor from which they originated. This is because of the selective pressure of the metastatic environment and the additional mutations accumulated during dissemination.

### 14.4. Epigenetic Heterogeneity

Epigenetics deals with modifications that affect gene expression without altering the DNA sequence [251]. These changes can be heritable and affect cellular behavior. In the tumor context, epigenetic alterations may play a crucial role in cancer genesis and progression.

**(a)** Main epigenetic mechanisms.

Epigenetic mechanisms in the initiation and development of cancers have a broad role [252], including the following:-*DNA methylation*: Adding methyl groups to specific regions of DNA (often in the promoter regions of genes) can repress their expression. In tumors, hypermethylation of tumor suppressor genes or hypomethylation of oncogenes may occur.-*Histone modifications*: Histones are proteins that DNA wraps around. Their modification (e.g., acetylation, phosphorylation, and methylation) can alter the structure of chromatin and the accessibility of DNA to transcription factors, affecting gene expression.-*Non-coding RNAs*: microRNAs (miRNAs) and other non-coding RNAs can regulate gene expression at the post-transcriptional level, and their alterations are often involved in tumor progression.-*Tumor plasticity*: Epigenetic alterations confer considerable phenotypic plasticity upon cancer cells, enabling adaptation to diverse microenvironments irrespective of genetic mutation. This plasticity can contribute to drug resistance and the ability to form metastases.

**(b)** Reversibility.

Unlike genetic mutations, many epigenetic alterations are reversible [253]. This makes the epigenetic “target” a promising area to develop new therapies.

-*Specific therapeutic adaptations for tumor heterogeneity*: Tumor heterogeneity requires personalized and dynamic therapeutic approaches. Table 5 illustrates the spectrum of these adaptations.

### 14.5. Advanced Diagnostics and Molecular Profiling

-*Liquid biopsy*: Allows the monitoring of the genetic and epigenetic alterations of the tumor in real time through the analysis of circulating tumor DNA (ctDNA) in the blood. This makes it possible to detect emerging new resistant clones or “targetable” mutations with no repeated invasive biopsies [254,255].-*Next-Generation Sequencing (NGS):* Allows the simultaneous sequencing of many genes or the entire genome/exome of the tumor, providing a detailed molecular profile of genetic alterations and tumor mutational burden (TMB). This can steer towards targeted therapies or immunotherapy [254].-*Multi-regional biopsies*: In some cases, especially for solid tumors, biopsies from different areas of the tumor may be necessary to capture the full range of intra-tumor heterogeneity [256].

### 14.6. Targeted Therapies

-*Molecularly targeted drugs:* These drugs inhibit specific proteins or signaling pathways that are altered in the tumor. Identifying “actionable” mutations (i.e., for which a specific drug exists) is crucial.-*Therapeutic combinations:* To counteract emerging resistance because of clonal heterogeneity, combination therapies that target multiple targets or target several cells subclones are being used. This approach reduces the likelihood that a single resistant clone can emerge and dominate.-*Cancer-agnostic therapies*: Several authorities have approved some drugs for specific mutations regardless of tumor location. For example, pembrolizumab treats tumors with high microsatellite instability (MSI-H) or mismatch repair deficiency (dMMR) [257], and larotrectinib treats malignancies with NTRK fusions [258].

Targeted therapies are drugs that act on specific “molecular targets” present in cancer cells, often because of genetic mutations. Here are some important examples. Chronic Myeloid Leukemia (CML) and Imatinib (Glivec) are one of the most emblematic examples and a real watershed in the history of oncology. Prior to Imatinib, CML was an often fatal disease with few effective treatment options. Imatinib is a tyrosine kinase inhibitor that blocks the activity of the BCR-ABL protein, a specific mutation present in almost all CML patients. Introducing Imatinib has turned CML into a chronic and manageable disease for most patients, with 10-year survival rates exceeding 90% [259]. Many patients can live normal lives by taking one pill a day.

Melanoma and BRAF/MEK inhibitors are important because about half of melanomas mutate in the BRAF gene (V600E) [260]. Drugs such as vemurafenib and dabrafenib (BRAF inhibitors) and trametinib, cobimetinib, and binimetinib (MEK inhibitors) have changed the outlook for patients with metastatic melanoma who have this mutation. The combined use of BRAF and MEK inhibitors has shown profound and durable tumor responses, with a significant increase in survival compared to traditional chemotherapy.

Non-small cell lung cancer (NSCLC) and EGFR/ALK inhibitors affect patients with NSCLC (those with adenocarcinoma and who are not smokers) who have mutations in the EGFR genes or rearrangements in the ALK gene [261]. Drugs such as gefitinib, erlotinib, and osimertinib (for EGFR) and crizotinib, alectinib, and brigatinib (for ALK) have been effective in these subgroups of patients. For example, patients with metastatic NSCLC with EGFR-sensitizing mutation treated with osimertinib have longer progression-free survival than those treated with chemotherapy [261].

A significant subset of NSCLC patients (10–15% in Caucasians and 30–40% in Asians) has activated mutations in the epidermal growth factor receptor (EGFR) gene. These mutations make the tumor sensitive to drugs called EGFR tyrosine kinase inhibitors (EGFR-TKIs). A complex problem of Spatial Heterogeneity (within the tumor) has been found in this cancer.

Even within the same primary tumor, not all cells may have the activating EGFR mutation. There may be subclones with different mutations or without EGFR mutations, which coexist [262]. Even more problematic is that, even before treatment, small populations of cells (subclones) that have already developed mechanisms of resistance to the future drug EGFR-TKI may be present. These “pre-existing” subclones may be too few in number to be detected by a single biopsy, but they can readily expand under the selective pressure of therapy. Unfortunately, researchers discovered a mechanism of temporal heterogeneity during the disease and treatment. Acquiring resistance is the most obvious manifestation of temporal heterogeneity [263]. Most patients with EGFR-mutated NSCLC respond very well to first-generation EGFR-TKIs (e.g., gefitinib and erlotinib) or second-generation EGFR-TKIs (e.g., afatinib and dacomitinib). However, almost all patients develop resistance to the drug within 9 to 14 months. The most common mechanism of resistance (occurring in about 50–60% of cases) is acquiring a secondary mutation in the EGFR gene, called T790M [264]. This mutation, which emerges only under drug pressure, alters the drug’s binding site on the EGFR, rendering first- and second-generation EGFR-TKIs ineffective. However, besides T790M, several other resistance mechanisms may emerge, often in different subclones. Different metastases can also develop different resistance mechanisms or have a different mutation profile. A biopsy from a single metastasis may not represent the entire heterogeneity of the patient.

Regarding Gastrointestinal Stromal Tumors (GISTs), Imatinib has also shown success for GISTs, a rare type of tumor that often has mutations in the KIT or PDGFRA genes [265]. Studies have shown Imatinib to improve survival and quality of life in these patients.

With T790M, one of the emerging diagnostics, liquid biopsies (circulating tumor DNA—ctDNA), is a particular issue [266]. It is a diagnostic technology that analyzes the DNA released by cancer cells into the blood (ctDNA). Unlike tissue biopsies, blood sampling is invasive and allows frequent repetition. This makes it possible to monitor the genetic evolution of the tumor in real time and to detect emerging new resistance mechanisms (such as T790M) long before the patient develops clinical symptoms of progression. ctDNA is a “pool” of DNA released from all tumor sites (primary tumor and metastases). This means that it can provide a more complete overview of the genetic heterogeneity of the tumor than a single tissue biopsy, which could sample only a part of the tumor or a single metastasis. Liquid biopsy has become essential to identify the T790M mutation in patients progressing on first/second-generation EGFR-TKIs. Detecting T790M in ctDNA allows for immediate osimertinib treatment, avoiding invasive tissue biopsies and therapy delays. After treatment, ctDNA can monitor minimal residual disease, detecting tumor DNA even when the disease is not visible with imaging techniques. This might one day inform decisions regarding additional therapies or the reintroduction of specific medications. However, after treatment, ctDNA can monitor minimal residual disease, detecting tumor DNA even when the disease is not visible with imaging techniques. This might one day influence decisions about additional therapies or the reintroduction of specific medications. In addition, liquid biopsy allows emerging new resistance mutations (even beyond T790M when patients progress on osimertinib) to be detected before progression is clinically clear. This paves the way for future sequential or combined therapeutic approaches.

An important consideration is how spatial and temporal heterogeneity in EGFR-mutated NSCLC is a prime example of how diversity within the tumor and its evolution under drug selective pressure can be the main cause of resistance and relapse [267]. Therefore, osimertinib, a drug designed to overcome the T790M mutation, and integrating liquid biopsies into clinical practice represent a significant “victory,” tailoring therapy based on the patient’s exact mutational profile, which can vary over time, and prolonging the duration of relapse. This illustrates how the detailed understanding of tumor heterogeneity is not just an academic exercise but a practical necessity to develop and apply emerging therapies that can improve patients’ lives.

### 14.7. Immunotherapy

-*Immune checkpoint inhibitors*: These drugs “unlock” the patient’s immune system, allowing it to recognize and attack cancer cells. The effectiveness of immunotherapy often depends on TMB (a higher TMB allows the immune system to recognize more neoantigens) and specific genetic alterations that affect the immune response.

A notable example is metastatic melanoma, which was the first tumor in which immunotherapy showed spectacular results. Drugs such as ipilimumab (anti-CTLA-4), nivolumab, and pembrolizumab (anti-PD-1) have revolutionized the treatment of metastatic melanoma. Patients who had a poor prognosis can now achieve complete and lasting remissions. Former president Jimmy Carter is a well-known example of successful immunotherapy for melanoma metastatic to the brain and liver, announcing that he is cancer-free after treatment with pembrolizumab.

Immunotherapy has also transformed the therapeutic landscape in non-small cell lung cancer (NSCLC). Patients whose tumors express high levels of PD-L1 often respond better to PD-1 inhibitors like pembrolizumab than to chemotherapy as a first-line treatment. This has led to a significant improvement in long-term survival for many patients. In kidney cancer (renal cell carcinoma), nivolumab and pembrolizumab, alone or combined with other agents, have shown excellent results in metastatic renal cell carcinoma, leading to high response rates and prolonging survival in a disease that was very difficult to treat. In Hodgkin lymphoma, patients with refractory or relapsed Hodgkin’s lymphoma have shown impressive responses to PD-1 inhibitors, with many achieving durable remissions even after failing other therapies. In tumors with high microsatellite instability (MSI-H) or mismatch repair deficiency (dMMR), a notable success is the “tumor type-agnostic” approval of pembrolizumab for all solid tumors with MSI-H or dMMR. A high mutational load is characteristic of these tumors, observed in colorectal but not limited to this location, resulting in a marked responsiveness to immunotherapy regardless of the primary site.

### 14.8. Epipharmaceuticals

-*Histone deacetylase inhibitors (HDACi):* These drugs act on histone modifications, altering gene expression and promoting cancer cell differentiation or death [268].-*DNA methyltransferase inhibitors (DNMTi):* These drugs can “reactivate” tumor suppressor genes silenced by DNA methylation [269]-*Combinations with standard therapies:* Clinicians combine epigenetic drugs with traditional chemotherapy or targeted therapies in studies to overcome resistance and improve response [270].

Epigenetic therapies are still a developing field, but they have already shown significant successes, especially in some hematological malignancies and when combined with other therapies [271]. In Myelodysplastic Syndromes (MDSs) and Acute Myeloid Leukemia (AML), physicians approve drugs, such as azacitidine and decitabine, which are inhibitors of DNA methyltransferases (DNMTi), to treat MDS and some forms of AML. These drugs can reactivate tumor suppressor genes that methylation has silenced, improving the survival and quality of life of these patients. Sometimes, they can also induce complete or partial remissions. Vorinostat, romidepsin, and belinostat, histone deacetylase inhibitors (HDACi), treat some cutaneous T cell lymphomas (CTCLs) and have received sanitary approval. These drugs work by modifying gene expression and can lead to durable responses in patient’s refractory to other therapies.

A recent Italian study (NIBIT-ML1) [272] has shown that combining epigenetic drugs with immunotherapy can help overcome resistance to immunotherapy in patients with advanced melanoma. This suggests a potential synergy between the two approaches and opens up new perspectives for patients who do not respond to immunotherapy alone. However, these examples show how understanding tumor heterogeneity at the molecular level has paved the way for much more precise and effective therapies, transforming the prognosis and quality of life of thousands of cancer patients. Research continues to identify new alterations and develop targeted drugs, hoping to extend these successes to an increasing number of cancers.

### 14.9. Dynamic Monitoring and Adaptation of Treatment

-*Molecular Tumor Board (MTB):* Establishing multidisciplinary teams composed of oncologists, pathologists, geneticists, and bioinformaticians is essential for interpreting complex molecular profiling data and guiding therapeutic decisions, considering the heterogeneity and evolution of the tumor [273].-*Track and trace approaches*: The idea is to monitor the tumor’s evolution (e.g., via serial liquid biopsy) and adapt therapy based on emerging new clones or resistance mechanisms [274].

In summary, tumor heterogeneity, both genetic and epigenetic, is an intrinsic feature of cancer that influences its behavior and response to therapies. Understanding this complexity is the basis of precision oncology, which aims to develop and apply personalized and dynamic diagnostic and therapeutic strategies to improve treatment efficacy and overcome resistance mechanisms.

After analyzing transmissible tumors based on genetic tumor traits possessed by parents, which involved issues related to genetic and epigenetic tumor heterogeneity and the therapeutic adaptations, such as immunotherapy and epigenetic drugs, it is important to analyze more closely the controversial and important hypothesis of tumor stem cells. It proposes and explains how a group of normal cells, the stem cells, can be at the origin of tumor heterogeneity.

## 15. Cancer Stem Cells: The Centrality of the Malignancy

We define cancer as an abnormal tissue growth because of uncontrolled cell division, which has an irregular lifespan and eventually generates a malignant tumor. The cancer stem cells (CSCs) are a small population of cells within a tumor that possess the ability to self-renew and differentiate into various cell types [275]. Mutated normal stem cells or dedifferentiated cells, which control the proliferative capacity of tumors, are believed to be their origin. But, unlike healthy cells, cancer stem cells continue to grow and produce new cells in an uncontrolled manner, leading to suppression of normal function with the potential progression to metastasis [275,276].

The cancer stem cell hypothesis proposes that the different cells in a heterogeneous tumor arise from a single cell, called a tumor stem cell [277]. The generation of CSCs arises from the transformation of somatic stem cells or differentiated cells, persisting as a tumor subpopulation while maintaining essential stem cell properties [275,276,277]. This hypothesis challenges the current understanding of cancer by introducing a novel model postulating that a small subset of tumor cells possesses unique self-renewal, differentiation, and tumorigenic capabilities. The capacity for lineage diversification (plasticity) is a key element of tumor heterogeneity [278]. Therefore, a heterogeneous population of partially differentiated cell types that resemble the cell types found in the tumor source tissue comprises tumors. This pluripotency and the self-renewal capabilities of CSCs allow them to differentiate into various tumor cell types with distinct characteristics, driving both inter-tumor and intra-tumor diversity [279]. The triple capacity of self-renewal, proliferation, and differentiation, characteristic of stem cells (SCs), could explain the patterns of both unrestrained and differentiated growth seen in benign and malignant tumors. CSCs may also adopt a relatively quiescent and slowly proliferating phenotype, resulting in decreased responsiveness to therapies targeting rapidly dividing cells.

The centrality of CSCs is a critical point in medicine and in cancer research. The role of these cells in the initiation of malignant disease, clonal expansion, and resistance to radiotherapy, chemotherapy, and immunotherapy underscores their significance [280]. It also highlights the importance of studying their homeostatic proliferation mechanisms, which are essential to their function and also become dysregulated during carcinogenesis. This suggests that cancer often results from a perversion of normal biological processes.

After years of searching for the “magic bullet” [281], the cure-all capable of destroying neoplastic cells with a targeted molecular mechanism, research is approaching a much more dynamic conception of cancer, in which the tumor is not a fixed target but changes continuously. It changes its cellular composition in space and time, reacts to therapies by becoming increasingly resistant, and actively interacts with all other cells, organs, tissues, and systems of the body, near and far. If CSCs are the key drivers, therapies that eliminate only most cancer cells will fail in the long term, requiring CSC-specific therapies. The concept of “stemness” as a spectrum implies that cancer cells can acquire stem-like properties, complicating targeting [282].

The quiescent and slowly proliferating phenotype of CSCs generates a critical limitation in cancer treatment because of its collateral damage to rapidly dividing healthy tissues. Namely, bone marrow, the gastrointestinal system, and skin exhibit heightened sensitivity to the effects of chemotherapy and radiation used in carcinoma treatment. Therefore, the role of CSCs in tumor initiation, drug resistance, metastasis, and recurrence underscores their importance as therapeutic targets. Table 6 compares the characteristics of normal cells and CSCs.

### 15.1. The Tumor Genesis and Clonal Expansion Guided by CSCs

CSCs are inherently “tumorigenic (tumor)” and can “generate tumors through stem cell processes of self-renewal and differentiation into multiple cell types” [283]. In neoplastic tissues, disruptions to the self-renewal mechanisms of cancer stem cells (CSCs) cause CSC overpopulation, promoting tumorigenesis and frequently increasing symmetrical divisions [284]. CSCs handle tumor initiation, progression, metastasis, and resistance to therapy [285]. They create a specific structure within a tissue called a niche, which is crucial for their functions [286]. This microenvironment controls the expansion and tumor-forming capacity of cancer stem cells (CSCs), showing the active contribution of the tumor microenvironment (TME) to CSC maintenance [286,287].

### 15.2. The Role of CSCs in Driving Tumor Heterogeneity

The cancer stem cell theory postulates that tumors are not simple monoclonal expansions of transformed cells but complex tissues where a minority, pathological pool of CSCs drives abnormal growth [288]. Thus, the intra-tumoral heterogeneity of tumors is driven by cancer stem cell differentiation into various cellular components. This function promotes cellular diversity within tumors, leading to differential responses to identical therapies among constituent cells. The pluripotency and self-renewal capabilities of CSCs allow differentiation into diverse tumor cell types, generating intra-tumoral and inter-tumoral heterogeneity [289].

### 15.3. Experimental Evidence Supporting the Tumorigenic Potential of CSCs

The first evidence came from the formation of colonies from tumor cells in vitro and in vivo. A significant finding was that a minority of tumor cells within each tumor exhibit tumorigenic potential upon transplantation into immunodeficient mice [290]. This minority rule explains why numerous tumors recur after initial treatment, as the CSC population remains [291]. Prospective studies have identified cancer stem cells (CSCs) in a variety of human solid cancers, such as breast, brain, colon, head, and neck, and pancreatic cancers [290,291]. As an illustration, in cases of colorectal cancer, tumor development typically requires the introduction of 200–500 EpCAM-high/CD44+ cells; in contrast, injection of 104 EpCAM-low/CD44− cells cannot induce tumor formation [292]. Conversely, tumors exhibiting a high EpCAM/CD44+ cellular profile maintained a differentiated phenotype, mirroring the morphological and phenotypic heterogeneity of the primary tumor mass [292].

### 15.4. Loss-of-Y Is a Stem Cell Mutation That Generates Cancer

Researchers observed a unique mutation, loss of the Y chromosome, in the cancer cells of some male patients [293]. They are entirely devoid of the Y chromosome. Chromosome aneuploidy in cancer cells is well known; however, the ramifications of this loss remain unclear [294]. Considerable data shows a role for stem cells in [295,296]. Studies show that this mutation may spread from cancerous cells to immune cells, compromising the latter’s anti-cancer capabilities because of Y chromosome loss [297,298]. The absence of the Y chromosome directly affects immune cells, rendering them inactive and immunosuppressive [299]. These losses can also compromise the efficacy of therapies, including CAR-T cell therapy, where researchers modify and reintroduce autologous immune cells to target the tumor [298,300].

The influence of stem cells on Y chromosome loss, achieved through multiple mechanisms, has significant implications, particularly in cancer. The depletion of the Y chromosome in stem cells, especially hematopoietic stem cells, has been correlated with heightened tumor malignancy and immune deficiency [301]. Y chromosome loss in tumor cells suggests a corresponding loss in immune cells. Concurrent depletion of Y in both hematopoietic and immune cell lineages correlates with highly aggressive tumorigenesis and immune deficiency, resulting in a poor prognosis [299,302]. Researchers studying Y loss have found a link between malignant and immune cells in the tumor microenvironment [297,303]. Another study has identified a gene on the Y chromosome as a possible therapeutic target in Acute Myeloid Leukemia [304]. However, the processes resulting in Y chromosome depletion within stem cells remain enigmatic. Genetic instability, defective DNA replication, and DNA damage accumulation may contribute to this.

The observation that tumors lacking Y chromosomes can be induced in immunocompetent mice possessing Y chromosomes [297,298,299] is supported by the concept of contagious transmission [301]. The neoplasms observed in these subjects exhibited an abundance of immune cells lacking the Y chromosome. This shows that immune cells enter the tumor in a healthy state, and a tumor microenvironment factor subsequently induces Y chromosome loss.

Losing the Y chromosome has been observed in patients with hematological diseases and sometimes also in healthy people, particularly older adults [301,302,305]. This loss may be associated with a higher probability of developing tumors with a very poor prognosis in older adults. The simultaneous loss of the Y chromosome in these types of cells could be associated with aggressive tumor cells, but also with a malfunctioning immune system.

To conclude, the Y chromosome plays a critical role in the immunological response to tumors. Thus, losing the Y chromosome in tumor and immune cells, a prevalent occurrence in older males, compromises the body’s disease-fighting capabilities, resulting in substantially reduced survival rates [306]. These studies may lead to the development of improved cancer therapies with enhanced efficacy in male patients.

### 15.5. Main Signaling Pathways in CSC Maintenance and Carcinogenesis

Accurate regulation of stem cell function is absolutely critical to normal biological activity, and their actions govern tissue development and homeostasis [307]. Tumors commonly over-activate many of these regulatory pathways, which are necessary for tumor growth. Thus, the dysregulation of proliferation pathways and differentiation, or the induction of oncoprotein activity, can transform a normal stem cell into a CSC [97]. Not surprisingly, cancer dysregulates many of the signaling pathways that contribute to the survival of CSCs [308]. Effectively, cancer hijacks existing developmental pathways. The interconnectedness and redundancy between these pathways mean that effective therapies targeting CSCs will probably require combination strategies. Table 7 shows the effects of cancer on regulatory pathways.

### 15.6. Cancer Stem Cells and Drug Resistance

CSCs contribute significantly to resistance to conventional therapies through various mechanisms [309,310].

(a)CSCs resist chemotherapy through several major mechanisms. Because of their quiescent proliferative state, these cells are unresponsive to therapies targeting rapidly dividing cells. They also activated drug efflux mechanisms. They exhibit overexpression of DNA repair mechanisms and anti-apoptotic genes. In addition, CSCs may secrete cytokines and chemokines, thus conferring therapy resistance upon other tumor cells.(b)CSCs are inherently more resistant to multiple clinical therapies, including radiation [311]. Radioresistance in these cells correlates with enhanced DNA repair mechanisms, robust reactive oxygen species (ROS) defenses, and inherent self-renewal capabilities. Compared with other methods, CRISPR-Cas systems show superior efficacy in mediating DNA repair and mitigating stress-induced DNA damage. Exposure to radiation may selectively eliminate radiosensitive tumor cells, while leaving radioresistant cancer stem cells (CSCs) viable; this selective repopulation from surviving CSCs contributes to adaptive radioresistance.(c)CSCs contribute to immunotherapy resistance through three main categories [97,312]:
-*Mechanisms related to cell surface proteins*: CSCs differentially express surface markers to escape immune surveillance and immune cell killing. This includes the downregulation of MHC Class I molecules, upregulation of CD47 (“Don’t Eat Me” signal [313]), and elevation of immune checkpoint ligands (e.g., PD-L1).-*Mechanisms related to cytokines released by CSCs*: CSCs can recruit immune cells and control immune responses by releasing pro-inflammatory cytokines (e.g., IL-1, IL-6, IL-8, and TGF-β) that impair anti-tumor immune responses and recruit immunosuppressive cells such as MDSCs and M2 macrophages.-*Mechanisms related to metabolic alterations*: CSCs show a significant production of glycolysis/lactate production. Lactate can activate CSCs, promote self-renewal, and induce an immunosuppressive phenotype in MDSCs and TAMs.

The breadth of mechanisms of resistance to CSCs shows that individual therapeutic approaches are unlikely to be fully effective, requiring multi-pronged strategies.

### 15.7. Current and Emerging Strategies for Targeting CSCs

Current and emerging strategies for targeting cancer stem cells (CSCs) [309] focus on different approaches that aim to eliminate or inhibit CSCs, which handle tumor progression, recurrence, and resistance to therapies. Cancer stem cells are a strategic therapeutic target in the treatment of tumors, aimed at reducing their proliferation and resistance to drugs. Current research is focusing on the tumor microenvironment to identify novel therapeutic targets for cancer stem cells, particularly in brain cancers [314].

Strategies include the use of antibodies, immunotherapy, CAR-T, modulation of quiescence, and resistance mechanisms, among others.

-*Antibodies:* Anti-OAcGD2 antibodies combined with TMZ (temozolomide) demonstrate efficacy in reducing tumor volume and the expression of CSC markers in GBM (glioblastoma multiforme), overcoming chemotherapy resistance.-*Immunotherapy:* Immunotherapy, including CAR-T therapy, aims to boost T cells to attack cancer cells, particularly CSCs, by exploiting the action of the immune system. CAR-T is a form of personalized medicine in which a patient’s genetically modified T cells express receptors that recognize specific tumor antigens.-*Radiotherapy:* Radiation therapy, including whole-body radiation therapy, can destroy cancerous cells, even CSCs, in particular blood cancers.-*Modulation of Quiescence*: CSCs can enter a state of quiescence or latency, which protects them from treatment and allows them to survive. Emerging strategies aim to modulate the mechanisms of quiescence to eliminate or inhibit CSCs in this state.-*Use of non-cancer stem cells:* Non-cancer stem cells, which are present in all organs and play a critical role in tissue repair, can treat several diseases, including some cancers, such as hematopoietic stem cell transplantation to treat leukemias and other blood cancers.

### 15.8. Importance of CSC Targeting

CSC targeting represents a key area of research to improve cancer therapies, prevent resistance and recurrence, and ultimately improve patients’ survival prospects [315,316]. The sections of greatest interest are

-*Resistance to therapies*: Researchers consider CSCs the major cause of resistance to traditional therapies, such as chemotherapy and radiotherapy;-*Tumor recurrence:* CSCs handle tumor recurrence after treatment and therefore are a key factor in tumor recurrence;-*Metastasis:* CSCs can migrate from the primary tumor and form metastases to other organs, contributing to the spread of the tumor.

## 16. Barriers Hindering Progress in Cancer Research

Several significant obstacles hinder progress in cancer research besides biological and technological constraints. The substantial financial investment required for cancer research, spanning basic science to clinical trials, presents a significant challenge [317]. Securing adequate and sustained funding from both public and private sectors, particularly for high-risk, high-reward research and the translational phase, remains a major hurdle [318]. The often-skewed distribution of research funding can also hinder the progress of young and emerging researchers with novel ideas [319]. The complex regulatory landscape governing cancer research, especially clinical trials, can be time-consuming and challenging. Stringent ethical considerations must always be paramount in research involving human participants [320]. Variability in regulatory processes across different countries can also complicate international collaborations [321]. Finally, the persistent and unacceptable disparities in cancer incidence, mortality, survival rates, and access to care across various population groups represent a major ethical and public health challenge [322]. Addressing these disparities requires a comprehensive understanding of the contributing factors and targeted interventions to promote health equity in cancer prevention, screening, treatment, and research.

## 17. Conclusions and Socio-Economic Aspects

All the above considerations show that cancer research and our biological understanding of this multifaceted disease are currently facing a complex web of interconnected limitations. These include the inadequacies of preclinical models in fully recapitulating human cancer,; the profound challenges posed by tumor heterogeneity at genetic, epigenetic, and phenotypic levels; the limitations in sensitivity, specificity, and accessibility of current early detection technologies; the intricate and often resistance-promoting influence of the tumor microenvironment; the ongoing quest for therapies that can selectively target cancer cells without harming healthy tissues; the pervasive problem of cancer cells developing resistance to treatment; and the significant gaps in our knowledge of the metastatic process.

Overarching these biological and technological hurdles are systemic barriers related to funding and resource allocation, regulatory and ethical considerations in research, and the persistent disparities in cancer research and care across different populations [323]. Overcoming these limitations will require sustained and collaborative efforts from researchers across diverse disciplines, policymakers, funding agencies, and patient advocates [323,324]. Continued innovation in preclinical modeling, advanced technologies for characterizing tumor heterogeneity and the microenvironment, the development of more precise and less invasive early detection methods, the discovery of novel therapeutic targets and strategies to enhance specificity and overcome drug resistance, and a deeper understanding of the mechanisms driving metastasis are all crucial for making significant strides against cancer. Addressing the systemic barriers through strategic funding initiatives, streamlined regulatory processes, and focused efforts to achieve health equity in cancer research and care is equally essential [325]. While the challenges are substantial, the ongoing progress and the promising directions for future research offer hope that we can continue to unravel the complexities of cancer and ultimately improve outcomes for patients worldwide. In conclusion, the problem is not only technological but also socio-political because the ability to manage, interpret, and integrate biomedical data is an organizational aspect that is still rather limited today compared to the complexity of tumor pathologies [323,325,326].

## Figures and Tables

**Table 1 cancers-17-02102-t001:** Impact of tumor heterogeneity on cancer therapy.

Level of Heterogeneity	Mechanisms Contributing to Heterogeneity	Implications for Targeted Therapy	Implications for Immunotherapy
Genetic	Mutations, genomic instability, exposure to mutagens.	Resistance because of lack of targets in subclones; outgrowth of resistant subclones with different mutations.	Variable antigen expression leading to immune evasion in some subclones.
Epigenetic	DNA methylation, histone modifications.	Resistance through altered expression of drug targets or resistance-conferring genes.	Variable expression of immune-related molecules.
Phenotypic	Genetic and epigenetic variations, TME interactions.	Differential drug sensitivity across subclones selection of drug-tolerant or resistant phenotypes.	Varying levels of immunogenicity; different interactions with immune cells; creation of immunosuppressive microenvironment by some subclones.

**Table 2 cancers-17-02102-t002:** Limitations and challenges in early cancer detection.

Screening/Detection Method	Key Limitations	Associated Challenges
Imaging (Mammography, CT, MRI)	Limited sensitivity for small tumors; not always cancer-specific; false positives; accessibility and cost.	Improving resolution and specificity; reducing false positives; increasing accessibility.
Tumor Markers (PSA, CA-125)	Poor accuracy and efficacy for many cancers; low sensitivity and specificity; false positives and negatives; non-cancerous conditions may raise levels.	Identifying more specific and sensitive markers; improving positive predictive value.
Multi-Omics	Ethical considerations on standardization of data interpretation and integration (data privacy).	Developing robust computational tools for data analysis and integration; establishing ethical guidelines.
Nanotechnology	Translation from lab to clinic, ensuring safety and efficacy in vivo.	Overcoming biological barriers for targeted delivery; long-term safety assessment.
AI and Machine Learning	Data quality and security; algorithm reliability and transparency; integration with existing systems; implementation costs; ethical and regulatory considerations.	Ensure explainability and fairness of algorithms; validate performance in diverse populations; establish regulatory frameworks.
Liquid Biopsies (ctDNA, etc.)	Low analyte concentration in early stages; need for highly sensitive and specific detection methods.	Improving detection sensitivity and specificity; distinguishing cancer-derived signals from background noise.

**Table 3 cancers-17-02102-t003:** Limitations of traditional preclinical cancer models and emerging alternatives.

Model Type	Key Advantages	Key Limitations
2D Cell Culture	Simple, inexpensive, high-throughput screening	Lacks 3D architecture, cell–cell/matrix interactions, complex microenvironment, immune component; limited clinical relevance.
3D Cell Culture	Improved structure over 2D; some cell–cell interactions	Often lacks vasculature, complex microenvironment, immune component; variability in protocols.
Murine Xenografts	Allows in vivo drug testing	Immunocompromised mice lack human immune system; murine microenvironment differs from human; limited metastasis in some models; physiological differences.
Humanized Mice	More relevant immune context for immunotherapy testing; can test unapproved drugs	Incomplete human immune system reconstitution; murine physiology still differs; expensive and technically complex.
PDXs	Preserves original tumor histology and genomics	Lacks fully intact human microenvironment (murine fibroblasts); expensive and difficult to generate; limited scalability.
Organoids	Better representation of human cancer heterogeneity; higher success rate than cell lines	Often lacks vasculature, complete microenvironment (stromal and immune components); need for standardized protocols.
GEMMs	Useful for studying cancer development driven by specific genetic alterations	Species-specific pharmacological and safety responses; time-consuming and expensive to generate and maintain.

**Table 4 cancers-17-02102-t004:** Genetic and molecular mechanisms favoring benign–malignant progression.

Genetic Mutations	Epigenetic Modifications	Chromosomal Abnormalities	Integration of Viruses and Mutagens
Mutations in oncogenes	DNA methylations	Deletions	Oncogenic Virus
Mutations in tumor genes	MicroRNA (miRNA)	Translocations	Chemical Agents and Radiation
Mutations in genes involved in DNA repair		Duplication	

**Table 5 cancers-17-02102-t005:** Therapeutic adaptations for tumor heterogeneity.

Advanced Diagnostics and Molecular Profiling	Targeted Therapies	Immunotherapy	Epipharmaceuticals	Dynamic Monitoring and Adaptation of the Treatment
Liquid biopsy	Molecularly targeted drugs	Immune checkpoint inhibitors	Histone deacetylase inhibitor (HDACi)	Molecular Tumor Board (MTB)
Next-Generation Sequencing (NGS)	Therapeutic combinations		Methyltransferase inhibitor (DNMTi)	Approaches “Track and Trace”
Multi-regional biopsies	Cancer-type agnostic therapies		Combinations with standard therapies	

**Table 6 cancers-17-02102-t006:** Comparative Characteristics of Normal Stem Cells and Cancer Stem Cells.

Property	Normal Stem Cells	Cancer Stem Cells
Self-renewal	Long-term ability to self-renew, maintaining tissue integrity.	Long-term self-renewal, but dysregulated, leading to overpopulation and tumor growth.
Differentiation Capacity	Differentiate into multiple specialized cell lineages for tissue function.	Differentiate into various cell types that make up the heterogeneous tumor.
Proliferative Potential	Some exhibit high proliferative potential, balanced for tissue regeneration.	High proliferative potential, essential for sustained tumor growth and mutation retention.
Homeostatic Regulation	Tightly regulated balance between self-renewal and differentiation, maintaining constant cell numbers.	Dysregulated self-renewal and impaired differentiation, leading to uncontrolled growth.
Genetic/Epigenetic Stability	Genetically and epigenetically stable.	Often carry genetic mutations and epigenetic changes that drive malignancy.
Tumorigenicity	Do not have tumor-starting ability.	Tumor-start (tumor-forming) and responsible for tumor genesis.

**Table 7 cancers-17-02102-t007:** Major signaling pathways in CSC maintenance and their dysregulation in cancer.

Pathway	Role in Normal Stem Cells	Dysregulation in CSCs/Carcinogenesis	Key Regulators/Components
Wnt	Essential for embryonic development, tissue homeostasis, and self-renewal of various adult stem cells (e.g., epidermis, intestine, and mammary gland).	Constitutive activation in many cancers; enhances CSC self-renewal, proliferation, invasion, and metastasis. Mutations in *APC* common in CRC.	β-catenin, Frizzled receptors, LRP5/6, APC.
Notch	Critical for cell fate specification, differentiation, and maintenance of stem/progenitor cells during development and in adult tissues (e.g., hematopoietic and neural).	Aberrant expression linked to poor prognosis; hyper-activation increases proliferation, angiogenesis, drug resistance, EMT, and BCSC numbers. Activating NOTCH1 mutations in T-ALL.	Notch receptors (1–4), Jagged/DLL ligands, NICD.
Hedgehog (HH)	Regulates cellular proliferation, differentiation, and migration during embryogenesis and in specific adult stem cell populations (e.g., neural and skin).	Commonly activated in cancer, promoting tumor progression and metastasis; associated with aggressive tumors and high CSC content. Mutations in PTCH1 predispose to medulloblastomas.	Sonic/Indian/Desert Hedgehog ligands, Patched (PTCH), Smoothened (SMO), Gli transcription factors (Gli1,2,3).
NF-KB	Involved in immune response, inflammation, survival, and differentiation; influences hematopoietic stem cell self-renewal.	Constitutive activation observed in many cancers; contributes to chemoresistance, tumorigenesis, and CSC self-renewal.	RelA, RelB, c-Rel, NFκB1, NFκB2, IKK.
HOW/STATE	Crucial for embryonic stem cell self-renewal, hematopoiesis, and neurogenesis.	Aberrant activation observed in CSCs from various tumors (e.g., breast, prostate, blood, and glial); promotes CSC proliferation and stemness.	JAK proteins, STAT proteins (e.g., STAT3).
PI3K/PTEN	Involved in cell cycle progression, growth, and survival; important for self-renewal in embryonic and hematopoietic stem cells.	Inactivating mutations in PTEN are common in glioblastoma; activation of PI3K/PKB or inactivation of PTEN leads to neoplastic phenotypes.	PI3K, PTEN, PKB (Akt).
Hippo	Regulates development, tissue homeostasis, and organ size.	Downstream effectors YAP/TAZ act as oncogenes, promoting proliferation, invasion, EMT, metastasis, and BCSC self-renewal.	MST1/2, LATS1/2, MOB1A/B, YAP, TAZ.
TGF β	Complex signaling network involved in normal and pathological processes, including cell growth, differentiation, and apoptosis.	Important regulator of tumorigenesis, inducing EMT and regulating CSC maintenance; correlates with poor prognosis.	TGF-β ligands, Type I/II receptors, SMAD proteins.

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
