# Peer review of "Overcoming Barriers in Cancer Biology Research: Current Limitations and Solutions"

_cancers, 2025, doi:10.3390/cancers17132102_

Round 1
Reviewer 1 Report
Comments and Suggestions for Authors
General considerations
The review is an elegantly written article, practically complete, on a relevant and current subject, which I consider necessary and timely to guide the current situation of cancer and its challenges in clarifying its etiopathogenesis and advancing knowledge to propose more efficient forms of treatment. The text is delightful to read, inviting the reader to explore current and interdisciplinary concepts regarding the multiple facets of malignant disease. The author develops the theme of the limitations of research in cancer biology, exploring the possibilities and barriers to overcome them in a conceptual and translational manner that is quite attractive to the reader seeking an update on this subject. The interdisciplinary approach the author takes in Appendix A, including the interface with information technology, quantum mechanics, and atomistic biology, is constructive for the reader to situate themselves in the deeper mechanisms that, in reality, occur at the heart of the genesis of malignant neoplasms. Furthermore, the author disseminates throughout the text several invitations for research projects on cancer, where questions have not yet been adequately answered or are controversial.
Specific considerations
In a merely allusive manner and without imposing compliance on the part of the author, I suggest that it would be of some interest to expand the content on the participation of cancer stem cells (SC) in the genesis of malignant disease, clonal expansion, and tumor resistance to radiotherapy, chemotherapy, and immunotherapy. The study of SC is of medical importance due to the homeostatic mechanisms of SC proliferation, which are the same processes that become deregulated in carcinogenesis. The discovery of these signaling pathways can improve treatments for neoplasias that are based on controlling SC proliferation. Additionally, the rapid division of tissues, such as bone marrow, intestine, and skin, is the first to be affected by chemotherapy and radiotherapy treatments for carcinoma. The three characteristics attributed to SCs—self-renewal, proliferation, and differentiation capacity—may explain both the unrestricted growth and differentiated growth patterns observed in both benign and malignant tumors. SCs share many properties with malignant cells, such as the ability to self-renew and proliferate, and carcinoma is believed to be a disease of SCs. Numerous arguments suggest that the origin of carcinoma may lie in stem cells.
First, SCs possess many of the characteristics that constitute the tumor phenotype, including self-renewal and essentially unlimited replicative potential. Second, the mutations that initiate tumor formation appear to accumulate in cells that persist throughout life, as suggested by the exponential increase in the incidence of carcinoma with age. This event reflects the requirement for four to seven mutations in a single cell to effect malignant transformation. Similarly, the formation of carcinoma from cells that persist throughout life is suggested by the increased incidence of skin tumors such as melanoma in adults after late childhood exposure to mutagens such as solar ultraviolet radiation. Normal somatic SCs are strong candidates for such persisting and neoplastic-initiating cells. Tumors themselves contain cancer stem cells. Tumors were once thought to be composed of cells, all of which had equal proliferative potential. However, there is evidence of a cellular hierarchy within some tumors, with cancer stem cells at the apex of this pyramid. In a heterogeneous population of cells, only cancer stem cells can self-renew, sustain, and increase the tumor cell population. Indeed, most cells within the tumor are incapable of independent growth and are readily susceptible to apoptosis. Only a small proportion of cells within the tumor are capable of independent development and meet the criteria for cancer SC. These cells have metastatic potential, form tumors in secondary hosts, and appear to be responsible for the continued renewal of cells within the tumor mass. These cells are most likely inclined to proliferate slowly and asymmetrically, self-renewing the SC population and giving rise to daughter cells that proliferate to sustain tumor growth. Conventional chemotherapy and antineoplastic radiotherapy target the daughter cells by dividing, altering the size of the tumor mass but leaving the cancer SCs intact, explaining the frequent rapid growth in tumor size once therapy is discontinued. Currently, one of the most exciting topics in oncology research is the identification of cancer stem cells and the exploration of their unique characteristics in the development of targeted therapies.
In addition, at the discretion of the author and editor, I believe that a textual addition on hereditary malignant neoplasms with a marked familial component could enhance the review, as these neoplasms may warrant a distinct approach in the context of the review article compared to sporadic tumors. Additionally, the author, at their discretion, could comment on any potential changes in the genetic or molecular machinery that could transform a benign neoplasm into a malignant one, given the current limitations of cancer biology research.
Author Response
See attached pdef

Reviewer 2 Report
Comments and Suggestions for Authors
The manuscript provides an ambitious review of the complex landscape of the limitations impeding progress in cancer biology research. It addresses critical areas such as tumor heterogeneity, inadequacies in preclinical models (2D/3D, xenografts, GEMMs), challenges in early detection, drug resistance, metastasis, and systemic barriers, including funding, regulation, and access, as well as emerging technologies. The breadth of the topics covered is a notable strength, offering a valuable overview of the field. However, this extensive scope compromises the depths of several crucial areas. The manuscript would benefit significantly from addressing conceptual gaps, providing more concrete evidence, refining its framing, and enhancing its clarity prior to publication.
Major Concerns: 1. Although numerous limitations have been identified, this type of analysis often remains superficial. Key concepts, such as "tumor heterogeneity, " are defined but lack a detailed exploration of how specific types (e.g., genetic versus epigenetic heterogeneity) necessitate distinct therapeutic adaptations. Similarly, discussions of epigenetic mechanisms and their clinical translation or therapeutic potential are underdeveloped. The "Emerging Technologies" section (multi-omics, AI, liquid biopsies, organ-on-chip, microfluidics) highlights potential but fails to adequately address the practical implementation challenges. Crucial aspects, such as computational demands, data interoperability, quality, security, required infrastructure investment, specialized training needs, and necessary policy or regulatory changes for widespread clinical adoption, are mentioned only briefly or superficially.
- The manuscript frequently proposes general strategies for overcoming barriers but provides insufficient concrete examples or case studies demonstrating where these strategies have successfully led to breakthroughs. For instance, discussions on overcoming preclinical model limitations lack specific examples of how organ-on-chip or advanced microfluidic systems have provided more physiologically relevant data, leading to improved clinical outcomes. Sections suggesting multidisciplinary collaboration require illustrative examples of successful collaboration, yielding tangible results. Recommendations for tackling systemic barriers (funding mechanisms and regulatory burdens) lack specific analysis (e.g., how current funding structures disadvantage high-risk or rare cancer research) or actionable policy proposals for streamlining identified burdensome processes.
- Intriguing concepts such as "deep molecular mechanisms" and the advocated shift towards network biology or proteoforms have been introduced but remain conceptually vague. The manuscript does not sufficiently clarify these ideas with concrete examples or address the significant practical difficulties in routinely mapping complex networks or proteoforms in clinical settings. Crucially, the link between achieving this "deep understanding" and generation of directly actionable therapeutic targets is underdeveloped and remains largely theoretical. This manuscript needs to better articulate the practical pathway from a complex mechanistic understanding to clinical application.
Minor Concerns: 1. Several grammatical errors, typographical mistakes, and ambiguous expressions were identified
2.Inconsistent terminology, such as the alternating use of "drug resistance" and "therapy resistance," diminishes clarity. Thus, a consistent terminology is crucial. Numerous sentences are excessively lengthy and complex, which impedes readability and comprehension. It is strongly recommended that these be divided into shorter, more focused sentences.
Recommendations for Revision:
- Increase Depth in Key Areas: Select to 2-3 critical limitations (e.g., addressing specific heterogeneity types, epigenetic therapy translation, and implementation challenges of a specific emerging technology) and provide a significantly deeper analysis, including specific examples and data where possible.
- Substantiate proposed strategies with specific case studies or published examples demonstrating successful implementation and impact. If space permits, add a table summarizing the key barriers, proposed solutions, and exemplar cases.
- Clarify Concepts and Translation Pathways: Provide clear definitions and concrete examples to elucidate "deep molecular mechanisms" and network/proteoform approaches. Explicitly discuss the practical steps and remaining challenges in translating this complex understanding into clinically actionable targets and therapies.
- Expand the discussion of challenges for emerging technologies (multi-omics, AI, and advanced models) to explicitly address infrastructure, cost, training, data management, and policy needs for real-world clinical integration.
- Remove or radically revise the concluding statement about the "one real barrier." This conclusion should synthesize the multiple complex barriers discussed without reductionist oversimplification. The argument in Appendix A is restructured to avoid circular reasoning regarding data validation and reliability.
Conclusion: This review addressed important and timely themes in cancer research. However, with its current breadth-over-depth approach, lack of specific evidence, conceptual vagueness, problematic framing, and clarity issues, it falls short of potential impact. Addressing the major and minor concerns outlined above, particularly by deepening critical analyses, providing concrete evidence, clarifying key concepts, correcting flawed arguments, and improving exposition, is essential for transforming this into a robust and valuable contribution to the literature. I recommend major revisions to the manuscript.
Author Response
see attached pdf

Round 2
Reviewer 2 Report
Comments and Suggestions for Authors
The author have worked on the comments, although it’s not fully explained the one’s I expected but it can be accepted now.